# To Spite the Devil: Martin Luther and Katharina von Bora's Wedding as Reform and Resistance

**Diane V. Bowers**

Pacific Lutheran Theological Seminary, Berkeley, CA, 94704, USA; dbowers@plts.edu

**Abstract:** The basis of Martin Luther's decision to marry Katharina von Bora on 13 June 1525, stemmed from his public, theological position that unless one were a particular exception, all men and women should marry. However, Luther's decision to marry when he did was controversial because the Peasants' Revolt raged, and it was surprising because up until November of 1524 Luther had stated that he was in his mind averse to marriage (for himself). Yet in May of 1525 Luther stated that he intended to take "his Katie" to wife, and in June, he did so. Why did Luther change his mind, and marry precisely when he did? I argue that the timing of his decision was influenced by Luther's apocalyptic sensibility that he was living in the last days, the immediate political context of the Peasants' Revolt, and the death of Elector Frederick the Wise of Saxony, Luther's patron and protector. The reason for his choice of Katharina von Bora as his wife included the need to secure for her financial support, but no less, her exercise of her own agency in choosing him as a husband. Can we say that Luther also personally warmed to the idea of marriage, drawn to the companionship of bed and table that it provided? There is certainly support for answering "yes." Luther's decision to marry was a theological, confessional, and political act, and yet these do not preclude the very human, personal, and relational factors in his decision.

**Keywords:** Martin Luther; Katharina von Bora; Reformation; reform; Wittenberg; marriage; wedding; Spalatin; Frederick; Peasant's Revolt; letters

## 1. Introduction

It is a popular and dramatic tableau in the imagination of Lutherans and historically-minded Christians, memorialized in story, art and film: the young monk Martin Luther stands before the entrance to the Castle Church in Wittenberg on All Saints Eve, 31 October 2017, and nails his "Disputation on the Power and Efficacy of Indulgences"[1], or the 95 Theses, to the church door. When upon publication the theses became an instant best seller, Luther's call to reform was carried throughout German speaking lands and beyond. This date is popularly designated as the beginning of the Lutheran or German Reformation.

---

[1] In 1968 Erwin Iserloh declared that Luther did not post the 95 theses to the Castle Church door in Wittenberg (Erwin Iserloh, *The Theses Were Not Posted: Luther Between Reform and Reformation*, trans. Jared Wicks (Boston: Beacon, 1968).) and currently there is not a scholarly consensus as to the historical legitimacy of the story. Those who argue that it was a historical event point out that it would have been in keeping with University custom. Philipp Melanchthon refers to posting the theses in his preface to the second volume of Luther's collected works (Volker Leppin and Timothy J. Wengert, "Sources for and against the Posting of the Ninety-Five Theses," Lutheran Quarterly 29 (2015), p. 374) and George Rörer, Luther's secretary, wrote in a note in 1540 that Luther posted the theses to the church doors of Wittenberg (Leppin and Wengert 2015, pp. 377–78). Leppin and Wengert's article is an excellent summary of the sources, and arguments for and against, the posting of the theses. We do know that Luther mailed the *Theses* enclosed with a letter to the Archbishop of Mainz, Albert of Brandenburg, on 31 October 1517, so in that sense one can fairly say that the theses were "posted".

Five years later, on Easter Eve, 4 April 1523, in an equally compelling but lesser-known action, twelve nuns, convinced by the teachings of the German reformation, risked their lives to escape from Marionthron Convent. The nuns had written to Martin Luther asking his help in obtaining their freedom, and Luther made the arrangements for them to escape hidden in the back of an empty delivery wagon. Nine of the twelve nuns were delivered to Martin Luther's care in Wittenberg, and one was Katharina von Bora, who two years later in 1525 later would become his wife. The Luther–von Bora marriage, which did not begin as a love-match, went on to become one of history's celebrated, happy and loving marriages.

Our image of the Luther marriage, based on letters Martin wrote to Katharina (unfortunately, all of Katharina's letters to Martin have been lost), and the observations of others, is consistently that of a loving, hardworking and engaged partnership in which Katharina managed a large household, raised animals and farmed, took in borders, and cared for the Luther's six children, supporting Martin in his work as a pastor, professor, and of course, leader of the Protestant Reformation. Their marriage and the way it became an ideal example of reformed marriage and family in general, almost seems inevitable, but it was not.

Although the Protestant Reformers and Luther himself were committed to the reform of marriage, Luther's decision to marry when he did was controversial in Protestant circles, primarily because of its proximity in time to the horrendous bloodshed of the Peasants' Revolt. In November of 1524 Luther wrote that he was in his mind averse to marriage, but in May of 1525 Luther stated that he intended to take "his Katie" to wife[2], and on June 12, he did so. Why did Luther change his mind, and marry precisely when he did?

The basis for Luther's decision to marry stemmed from his public, theological position that unless one were a particular exception, all men and women should marry. Just as he encouraged his colleagues and other public figures to marry, he was called upon to set the example. Part and parcel of the reformers' advocacy for the freedom of clergy to marry was their desire to reform marriage as an institution: to make it easier for all to marry by reforming marriage laws, and to raise the state of marriage to a position of honor and a calling.

However, once Luther decided he would marry, he married quickly, during circumstances that many thought made his choice insensitive, to a woman whose status as a former nun would only increase the virulence of the Roman Catholic response. The well-documented motive for Luther's decision to marry Katharina von Bora in particular was his commitment to find a husband for her and secure her financial support.

In addition, I argue that the timing of his decision was influenced by Luther's apocalyptic sensibility that he was living in the last days, and his own sense of personal danger in the immediate political context of the Peasants' Revolt. A somewhat lesser known, contributing factor was the death of Elector Frederick the Wise of Saxony, Luther's patron and protector.

But can we say that Luther also personally warmed to the idea of marriage, drawn to the companionship of bed and table that it provided? There is certainly support for answering "yes", and to argue that it is more than conjecture, but a reasonable possibility that Luther married in part because he cared for von Bora, and that von Bora's choice of Luther as a husband was an exercise of her own agency, are feminist claims.

In this paper I will expound upon the above factors, and look closely at letters written by and to Luther in 1524 and 1525, to trace the events that impacted Luther's decision to marry, and his own reflection on and interpretation of it.

---

2    Luther, Martin. 1918. *Luther's Correspondence and Other Contemporary Letters*. Translated and Edited by Preserved Smith and Charles M Jacobs. Philadelphia: The Lutheran Publication Society. Vol. II, 1521–1530, let. 667, pp. 309–10. Hereafter referred to as Luther 1918.

## 2. The Theological and Social Reform of Marriage

The Roman Catholic Church of the early sixteenth century recognized that marriage was a natural institution, a gift of God, necessary for procreation, and called it not only an antidote to the sin of lust, but a sacrament. As a sacrament, this meant that marriage conveyed grace not only to the couple but to society. Yet the church simultaneously made marriage difficult by imposing impediments to marriage, and the church upheld the moral superiority of virginity and the contemplative life to that of marriage.

The very complexity and prohibitive nature of marriage laws deterred or prevented many from marrying. The extensive nature of the prohibitions imposed by consanguinity (related by blood) and affinity (spiritual relationships such as being a god-parent or god-child) laws, sometimes to the seventh degree, made it difficult if not impossible for the average sixteenth century peasant, living in an ancestral village, a half day or day's journey from the next village, to even find, let alone court and win, a legally acceptable marriage partner. Finding suitable spouses was difficult for noble families as well, who had much greater resources.

Sacramental marriage laws were in tension with the concept of marriage as a natural institution[3]. By supporting the ideal of virginity and the taking of vows as preferable to marriage, the church's message that marriage was a natural state and a duty and remedy for sinful people was accompanied by a second message which said that marriage was most appropriate for the weak, those unable to control their base sexual nature, and that the life of the celibate priest, nun, or monk, was superior and more pleasing to God. The Protestant reformers recognized these and other tensions, exploited their weaknesses in support of reform, and set about to change and restore the state of marriage[4].

What sort of self-imposed task faced the Protestant reformers? Women were popularly portrayed as being ruled by their carnal natures and as temptresses seeking to ensnare men with their sexual wiles. Marriage was a trap, an eternal arrangement in which the husband was chained to a wife and a pack of children who drained his time, energy, patience and wallet[5]. In 1541 reformer Sebastian Franck described a "rampant state of divorcing or running away . . . where one partner abandons the other in exigencies just when they need each other most"[6].

In reality, forty percent of all women were single. Twenty percent never married, and ten to twenty percent were widows. Infant and child mortality was high: one third to one half of all children died by age five. Women were physically vulnerable to violence, including as the main targets of both

---

3    (Witte 1986, p. 307).
4    Did the Reformation, or the reform of marriage specifically, change the lives of women in ways that can be viewed as positive or liberating? This tremendously important question arises naturally when considering the reformers' views of marriage and the life of Katharine von Bora, as we are doing in this paper.The issue is complicated. For example, reformers elevated the theological understanding of the state of marriage as well as the role of mother and housewife to that of a calling. At the same time, "The theologically elevated role of mothers did not entail improved political or legal equality for women". (Kirsi Stjerna, *Women and the Reformation*, Blackwell Publishing: Oxford, p. 220). Within Protestant territories, convents—offering one of few socially acceptable roles available to women other than marriage—were closed. So, did the Reformation bring about positive changes in the lives of women? The answer is that it depends.It depends on whether the woman in question was personally persuaded by Reformation teaching or not; whether she accepted gladly the role of wife and mother; indeed, upon her sexual orientation (the concept of sexual orientation is a modern one; a woman at the time would only have known that she did not feel towards men as she supposed all other women felt). As convents were closed in Reformation territories, the Reformation was good news for women who had been forced into convents and bad news for those who wished to remain there, especially for those (wealthy) women who found in the convents access to education and a level of independence and even authority.In her important assessment, Kirsi Stjerna points out that not only was it not the goal of the reformers to overturn patriarchal systems or to achieve liberation or autonomy for women, but the Reformation needed the continuity provided by hierarchical, gendered relationships, in order to succeed (215). However, Stjerna points out, despite their restricted roles, the women of the 16th century should not be viewed only as victims, passive, or unthinking. Quite the contrary, women too "were fueled by the reform ideas, and contributed in their own ways within the existing structures" (222). In particular, women married to Protestant pastors had roles of elevated, although domestic, importance within the community. There is no better example of this than Katharina von Bora.
5    For an excellent overview see (Ozment 1983).
6    (Hendrix 1992, p. 254).



secular and ecclesiastical witch-hunters. Eighty percent of the one hundred thousand people executed for witchcraft between 1400 and 1700 were women, mostly older and single[7].

While without statistics ready at hand, the reformers exhibited an awareness of women's vulnerable state in society and a strong commitment to their worth as God's creation. In his tract on *The Estate of Marriage*, Luther identifies a cultural antagonism towards both women and marriage, and condemns it. He writes,

> What we would speak most of is the fact that the estate of marriage has universally fallen into such awful disrepute. There are many pagan books which treat of nothing but the depravity of womankind and the unhappiness of the estate of marriage, such that some have thought that even if Wisdom itself were a woman one should not marry ... So they concluded that woman is a necessary evil, and that no household can be without such an evil. These are the words of blind heathen, who are ignorant of the fact that man and woman are God's creation[8].

In *The Babylonian Captivity of the Church*, published in 1520, Luther argued that marriage was never "divinely instituted" as a sacrament, and railed against canonical impediments to marriage[9]. These included "spiritual affinities"[10], for example, relationships that arose through sponsorship at baptism, and the "impediment of public decency"[11] which held that no relative of a deceased fiancé could marry the fiancé, even to the fourth degree of consanguinity, and many more.

Luther's 1522 treatise, *On the Estate of Marriage*, is remarkable for its theological statements about human nature. In it Luther argues that, apart from those who either physically cannot engage in sexual intercourse or have been given the gift of celibacy (and these are rare, Luther says, "not one in a thousand"), every person should marry, for marriage is natural and more necessary than sleeping, waking, eating, drinking, and emptying the bowels and bladder. It is a "disposition just as innate as the organs involved in it"[12].

Citing Genesis 1, verses 27 and 28, Luther writes that just as it is not within a man's power to not be a man or a woman's power to not be a woman (" ... male and female God created them"), so it is no one's prerogative to be without the other ("Be fruitful and multiply ... "). "It is not a matter of free choice or decision but a natural and necessary thing, that whatever is a man must have a woman and whatever is a woman must have a man"[13]. The surprising element in this sermon, "one still highly offensive in the sixteenth century—was the assertion that sexual drives were a divine force or even God's vital presence"[14].

## 3. Luther's Decision to Marry

Luther, who recommended marriage to so many, did not marry until the age of 42, and when he did, the marriage appeared sudden and ill-timed to many, was welcomed with delight by his parents, was a surprise to his friends and colleagues, and a scandalous outrage to his Roman Catholic opponents. Happily, Luther's marriage to Katharina von Bora in June of 1525, undertaken as a theological statement and act of protest, became a happy and loving marriage that served the cause of the Reformation in many ways, not least of which was by increasing Luther's physical and spiritual well-being and length of life.

---

7    (Ozment 1983, p. 1).
8    (Luther 1955–1986, vol. 45, p. 36).
9    (Luther 2016).
10   Ibid., p. 104.
11   Ibid., p. 106.
12   (Luther 1955–1986, vol. 45, p. 18).
13   Ibid., p. 18.
14   (Oberman 1989, p. 273).

Before the year 1525, Luther stated more than once that he himself did not intend to marry even while advocating that others should do it. On 30 November 1524, Luther wrote a letter to George Spalatin, ordained priest, chaplain and secretary to Elector Frederick III of Saxony, and his good friend, counseling Spalatin as to whether Spalatin might marry, and then turned the subject to himself.

> I thank Argula[15] for what she wrote me about marrying, nor do I wonder at such gossip when so many other things are said about me; please give her my thanks and say that I am in the hand of God, a creature whose heart He may change and rechange, may kill and make alive, at any hour or minute, but that hitherto I have not been, and am not now inclined to take a wife. Not that I lack the feelings of a man (for I am neither wood nor stone), but my mind is averse to marriage because I daily expect the death decreed to the heretic[16].

Luther's position is clear: although he was not a man of stone, he was not inclined to take a wife because he expected to soon die the death of one declared a heretic.

The November 30 letter also includes Luther's response to Spalatin's consideration of whether, due to various frustrations, he should resign from his position at the court. Luther reminded Spalatin of his usefulness to the court and offered his opinion that Spalatin should stay, but said that if he was going to leave his position at the court in order to marry, there was no point in keeping it a secret.

Luther wrote, "I cannot imagine what other reason you have for wishing to resign, unless your desire to marry weighs on you … Therefore remain, or, if you must go, let it be for no less a reason than to take a wife. Perhaps you fear to give this as a reason, and prefer some excuse in its place—a course of conduct of which I fail to see the advantage, as the real cause would come out as soon as you married"[17]. If Luther thought that Spalatin might be looking for an excuse to leave his position with the court in order to marry, it is reasonable to think that Frederick's opposition to the marriage of clergy was the reason. Although a staunch defender of Luther, Frederick remained conservative on some matters, including the marriage of priests[18].

Frederick's opposition to clergy marriage, and then his death on 5 May 1525, may well have been a reason for Luther's delay of marriage and then the suddenness of that marriage. Luther was aware of the Elector's failing health, counseling Spalatin on 2 December 1524, not to "desert the Elector when he is perhaps very near the grave … "[19] Luther married von Bora on June 13, a little more than a month after the Elector's death in May. Lull and Nelson suggest that, "Perhaps Luther felt a freer hand to proceed with this towering but controlling figure out of the way"[20].

However, Luther's decision to marry soon after the Elector's death may also have reflected Luther's increasing spiritual authority. After Frederick's death Luther sought to more firmly establish the connection between Protestant identity and princely rule with the support of Frederick's brother and successor, Elector John. By Spalatin's invitation Luther was instrumental in planning Frederick's carefully choreographed and very public funeral and burial in Wittenberg on 10–11 May 1525, and Luther made sure that the ritual intertwined Protestant theology and courtly authority. For example, language suggesting that prayers for the dead might influence the soul's fate was stricken, and the casket, which remained overnight in the Castle Church, was bedecked with Frederick's coat of arms[21].

---

[15] Luther is referring to Argula von Grumbach, nee Stauff, of Bavaria, who, though relatively unknown today, was a powerful presence in the early years of the Reformation. Educated, highly intelligent, and a theologian, von Grumbach wrote and effectively harnessed the printing press to publish pamphlets in defense of the Protestant cause. She corresponded with Melanchthon, Luther and Spalatin. Here it seems that von Grumbach had written to Luther encouraging him to marry, and he references that letter.

[16] (Luther 1918, let. 648, p. 264).

[17] Ibid., let. 648, pp. 263–64

[18] (Brecht 1990).

[19] (Luther 1918, let. 649, p. 264).

[20] (Lull and Nelson 2015).

[21] (Krentz 2014).

Frederick's protection of the reformers at Wittenberg and support of Evangelical reforms in his territory, as well as his financial support of Luther personally, were essential. If Luther had chosen to marry while Frederick lived, no one can say whether this would have caused a fundamental rift between them, but Luther may have found it the greater part of wisdom to refrain from causing the Elector undue aggravation. However, Luther's quiet wedding a month after the funeral and the public celebration and feast two weeks after that may have reflected twin freedoms that Luther now felt—the freedom to marry, which Luther strongly proclaimed was God-given, and this as part of his emerging freedom to consolidate Reformation gains by establishing Protestant worship throughout the electoral territory of Saxony[22].

By the spring of 1525 clearly Luther was thinking of marriage, and Katharina von Bora's name appears in his correspondence with Spalatin. In a letter dated 16 April 1525, Luther responds to Spalatin's suggestion that he marry, and we hear Luther urging Spalatin to "carry out your intention".

> You write about my marrying. You ought not to wonder that I, who am such a famous lover, do not take a wife; it is more wonderful that I, who write so often about matrimony, and thus have so much to do with women, have not long since become a woman, to say nothing of marrying one. But if you wish me to set you an example, you already have a great one. For I had three wives at the same time, and loved them so bravely that I lost two of them, who are about to accept other wooers. The third I am only holding with the left arm, and she, too, perhaps, will soon be snatched away from me. But you are such a laggard in love that you do not venture to become the husband even of one woman. But look out, or I, who have no thought at all of marriage, may sometime get ahead of you prospective bridegrooms. It is God's way, to bring to pass the things you do not hope for. I say this that, without jesting, I may urge you to carry out your intention[23].

The three women whom Luther jokingly referred to as his wives were probably the sisters Ave and Margaret von Schönfeld, and Katharina von Bora. These three were the last of the women who had fled the Marionthron Convent in Nimbschen and remained unmarried.

In a pleasing coda to the ongoing conversation about marriage between the two friends, six months after Luther's marriage in June 1525, Spalatin married a woman whose name was also Katharina, in December. Luther was highly pleased by Spalatin's marriage as well as the coincidence of their wives' names. He wrote a letter to Spalatin dated December 6, shortly before the wedding, expressing his delight on behalf of his good friend. "I wish you grace and peace in the Lord, and also joy with your sweetest little wife, also in the Lord. Your marriage is as pleasing to me as it is displeasing to those priests of Baal. Indeed, God has given me no greater happiness, except the Gospel, than to see you married, though this, too, is a gift of the Gospel, and no small fruit of our evangelical teaching"[24].

Luther continued his letter explaining that he would not be able to attend the wedding, and joked that, poor as he was, he would have sent Spalatin the Portuguese gold piece that he had given Katharina, if he did not fear that Spalatin would be offended. The letter concludes with a fond, romantic, and even erotic, blessing, in the words of a man enjoying the delights of the marriage bed. "When you have your Catharine in bed, sweetly embracing and kissing her, think: Lo, this being, the best little creation of God, has been given me by Christ, to whom be glory and honor. I will guess the day on which you receive this letter and that night I will love my wife in memory of you with the same act, and thus return you like for like. My rib and I send greetings to you and your rib"[25]. Oberman's translation is more pleasing: "On the evening of the day on which, according to my calculations, you

---

22 On the establishment of Protestant worship in Saxony see (Krentz 2014, pp. 327–29).
23 (Luther 1918, let. 672, pp. 305–6).
24 Ibid., let. 719, p. 355.
25 Ibid., let. 719, p. 356.

will receive this, I shall make love to my Catherine while you make love to yours, and thus we will be united in love"[26].

## 4. The Peasant's Revolt and Bad Timing

Luther's friends and colleagues, and in particular Philipp Melanchthon, had reservations about the timing of Luther's marriage on June 13 of 1525 because it happened during the horrific events of the Peasants' War. Germany was reeling from the violence of recent months, and so a wedding seemed inappropriately light-hearted, and a distraction when Luther's leadership was needed. Some feared that the incendiary nature of his wedding, particularly because Luther was a monk marrying a nun, which was considered spiritual incest[27], and the attacks that would follow on his character, would damage the cause of the Reformation.

In a letter written in Greek to his friend Joachim Camerarius on 16 June 1525, three days after Luther's wedding Melanchthon poured out his concerns. The letter is important to those interested in the significance of Luther's wedding because it is a personal and immediate response from one in Luther's inner circle. It was also used extensively by Roman Catholic writers to condemn Luther as driven by lust, and is worth a long excerpt here. Melanchthon wrote,

> Greetings. Since dissimilar reports concerning the marriage of Luther will reach you, I have thought it well to give you my opinions of him. On June 13, Luther unexpectedly and without informing in advance any of his friends of what he was doing, married Bora; but in the evening, after having invited to supper none but Pomeranus and Lucas the painter and Apel[28], observed the customary marriage rites. You might be amazed that at this unfortunate time, when good and excellent men everywhere are in distress, he not only does not sympathize with them, but, as it seems, rather waxes wanton and diminishes his reputation, just when Germany has especial need of his judgment and authority.
>
> These things have occurred, I think, somewhat in this way: The man is certainly pliable; and the nuns have used their arts against him most successfully; thus probably society with the nuns has softened or even inflamed this noble and high-spirited man. In this way he seems to have fallen into this untimely change of life. The rumor, however, that he had previously dishonored her is manifestly a lie. Now that the deed is done, we must not take it too hard, or reproach him; for I think, indeed, that he was compelled by nature to marry. The mode of life, too, while, indeed, humble, is, nevertheless, holy and more pleasing to God than celibacy.
>
> When I see Luther in low spirits and disturbed about his change of life, I make my best efforts to console him kindly, since he has done nothing that seems to me worthy of censure or incapable of defense. Besides this, I have unmistakable evidence of his godliness, so that for me to condemn him is impossible. I would pray rather that he should be humbled than exalted and lifted up, as this is perilous not only for those in the priesthood, but also for all men. ... Besides, I have hopes that this state of life may sober him down, so that he will discard the low buffoonery which we have often censured[29].

It worth noting Melanchthon's claim that Luther's wedding (if not his intention) was a complete surprise to all his friends. He expressed his dismay at the timing of the event, given that all of Germany was in distress. Melanchthon suspected what Luther's Catholic critics would roundly proclaim that Luther had succumbed to the nuns' arts (Luther himself would say this was not so) but he immediately

---

[26]  (Oberman 1989, p. 276).
[27]  (Fudge 2003).
[28]  Melanchthon is referring to Johannes Bugenhagen, Lucas Cranach, and John Apel.
[29]  (Luther 1918, let. 692, p. 325).

moved to defend his friend, declaring the rumor that they had slept together before the wedding a lie. Then he looked to bright side of the situation, stating that marriage is a more honorable mode of life than celibacy, and even offered his hopes that the marriage would settle Luther down a little, causing him to discard "the low buffoonery" that the others had so often had to censure.

All in all, it is a remarkable letter, capturing the difference between Luther's impetuous bravery and Melanchthon's greater cautiousness; the political situation of the German nation as well as the rumors and whispers surrounding the wedding; Reformation theology regarding marriage and celibacy; and the two men's friendship, as Melanchthon rallied to Luther's side and hoped for the best.

Oberman states that the Peasant's War precipitated Luther's decision to marry, writing that while Luther was well aware of the ways in which the common people were oppressed by the authorities, he considered "violent resistance against the authorities, particularly in the name of the Gospel" to be "the work of the Devil—and only possible in the chaos of the last days"[30]. According to Oberman, Luther believed that the end was near, that the devil roamed the land, and that the devil hated the joys of marriage as much as the good news of the gospel. Therefore, Luther determined to marry Katharina to "spite the devil".

The letter Oberman cites above, and from which the title of this paper is taken, written by Luther to John Rühel on 4 May 1525, while Luther was traveling, is important and revealing. The letter is mostly taken up with Luther's condemnation of the Peasants' War that had spread to the north and was threatening the rule of the Saxon and Thuringian princes. However, the letter reveals two other things. First, that Luther's thinking about the possibility of his own marriage had evolved and he was now planning to marry Katharine von Bora.

Second, it indicates that whereas previously, the danger Luther constantly found himself in was a hindrance to marrying, now, the imminent danger of the Peasant's War had the opposite effect. I believe that his decision was hastened, not only by his general apocalyptic sensibility that he was living in the last days, but by the specific possibility of the violence of the marauding peasants reaching Wittenberg. Luther wrote,

> This matter [the Peasants' revolt] concerns me deeply, for the devil wishes to kill me. I see that he is angry that hitherto he has been able to accomplish nothing either by fraud or force; he thinks that if he were only free of me he could do as he liked and confound the whole world together, so I almost believe that I am the cause that the devil can do such things in the world, whereby God punishes it. Well, if I can get home I shall meet my death with God's aid, and await my new masters, the murderers and robbers who tell me they will harm no one. Highway robbers always say the same: "I will do you no harm, but give me all you have or you shall die". Beautiful innocence! How fairly the devil decks himself and his murderers! Before I would yield and say what they want, I would lose my head a hundred times, God granting me His grace. If I can do it before I die, I will yet take my Katie to wife to spite the devil, when I hear that they are after me. I hope they will not take away my joy and good spirits[31].

The Peasants' War had apocalyptic meaning for Luther. To him the upheavals meant the world was ending[32]. We see this in Luther's treatise, *Against the Robbing and Murdering Hordes*, written in the spring of 1525, which ends as follows: "To this let every pious Christian say, 'Amen.' For this prayer is right and good, and pleases God; this I know. If anyone thinks this is too harsh, let him remember that rebellion is intolerable and that the destruction of the world is to be expected every hour"[33]. Therefore,

---

[30]　(Oberman 1989, pp. 277–78).
[31]　(Luther 1918, let. 667, pp. 309–10). Again, Oberman's translation of the last sentence is more pleasing: "I will not be robbed of my heart and my happiness". p. 278.
[32]　(Lull and Nelson 2015, p. 241).
[33]　(Luther 1955–1986, LW 46, p. 55). Cited by (Lull and Nelson 2015, p. 241).

Luther hurried back to Wittenberg to marry Katharina before the peasants arrived to murder him, or the world ended, whichever came first.

## 5. Katherina von Bora

While not as famous as the story of Luther nailing 95 theses to the church door, the escape of twelve nuns, hidden in a covered delivery wagon from a convent near Grimma in 1523, and the arrival of nine of them in Wittenberg, is both vivid and heroic. How did it come to pass that 24 year old Katharina von Bora found herself fleeing through the night hidden in a wagon, endangering her life for the sake of the gospel, and then two years later married to the leader of the Protestant Reformation, Martin Luther?

Katharina von Bora was born on 29 January 1499, to Hans von Bora and Katharina (or Anna) von Haubitz (or Haugwitz), a moderately noble family whose fortunes were at an ebb. For this reason, perhaps, and because her mother died and her father remarried, she was sent to the Cistercian cloister of Marienthron, at Nimbschen, in 1509 at the age of 10, and took her vows at the age of 16.

In 1519, Luther preached in the town of Grimma, near Nimbschen. The prior at the Augustinian monastery at Grimma had two nieces at Marienthron, and Luther's ideas flooded the convent[34]. By 1523, Katharina and eleven others wanted to leave, including Magdalena von Staupitz, the niece of John Staupitz, Luther's father confessor in the Augustinian order. Letters were smuggled out of the convent to Luther, and Luther made arrangements.

On Easter Eve, 4 April 1523, Leonhard Koppe, a 60 year old town councilor and tradesman and from Torgau, made a delivery to the convent of food items, and smuggled twelve nuns out, hidden in his wagon[35]. While the image of such a journey is humorous, the reality was not. Herr Koppe and the nuns risked their lives, traveling through the territory of Duke George to the safety of Torgau, which was in the territory of Luther's protector, Duke Frederick. Freeing or kidnapping nuns was against the law and Duke George had previously executed a man for such a crime.

Fortunately, Koppe and his charges arrived safely in Torgau. Three of the women returned to their families, and after observing the Easter services, on Easter Monday Koppe transported the remaining nine nuns to Wittenberg and left them in Luther's care. Luther wrote to Spalatin on Easter Wednesday, April 10, describing the helplessness and destitution of the women, and asking for assistance.

"Nine fugitive nuns", Luther wrote, "a wretched crowd, have been brought me by honest citizens of Torgau. I mean Leonard Coppe and his nephew and Wolf Dommitzsch; there is therefore no cause for suspicion. I pity them much", but not so much as those who remain trapped in the cloisters in their "cursed and impure celibacy". Luther planned to ask their families to provide for them, and to provide for them himself if their families could not. The women were Magdalene von Staupitz, Elsa von Canitz, Ave Gross, Ave von Schönfeld and her sister Margaret, Laneta von Goltz, Margaret and Catharine Zeschau and Catharine von Bora.[36] Luther asked Spalatin to beg for money from his rich courtiers to help support these women[37].

The next day, 11 April 1523, a letter from Dr. Nicholas von Amsdorf to Spalatin clarified that twelve nuns had left the convent but nine had arrived in Wittenberg (there had been some question). Amsdorf humorously declared to Spalatin that he had designated the highest born of the women [Magdalene von Staupitz] to be Spalatin's wife, and then asked again for assistance, saying, "If you want to give anything to the poor, give it to them, for they are poor, wretched, and deserted by their

---

34 (Smith 1999, p. 746).
35 The popular story is that the nuns were hidden in herring barrels, but what is known is that Koppe delivered herring to the convent. The nuns were more likely hidden in the covered wagon as if they were empty barrels.
36 The sharp-eyed reader will have noticed "Catharine" here, whereas elsewhere in the paper she is "Katharina" von Bora. When referencing letters translated by Preserved Smith, I follow the spelling of his translation. Elsewhere I use "Katharina," which is now standard.
37 (Luther 1918, let. 583, pp. 179–80).

kinfolk. I pity the poor things; they have neither shoes nor clothes. I beg you, dear brother, to see if you cannot get something for them from the people of the court, so that they may be provided with food and clothing. Please do all you can, for in their great poverty and anxiety they are very patient. Indeed I am astonished that in such great tribulation and poverty they are so patient and happy"[38].

The observations offered by both Luther and Amsdorf highlight the women's precarious position. They had been unable to bring with them in their flight from the convent even extra clothes or shoes, and having escaped, they had no means of support if their families would not take them back. These nine nuns were not the only fugitives to come to Luther's door. In a 1523 letter, Luther complained that if he did not have to "waste so much money on runaway monks and nuns", he would have been able to provide help to others[39].

In the following days Luther sought residences for the women and also husbands. Katharina moved into the home of the painter Lucas Cranach and his wife, where she probably worked as a serving maid. In 1523 a young man named Jerome Baumgärtner from a well-to-do family in Nuremberg, who had previously studied at Wittenberg, visited again, and a romance developed between him and Katharina. Her expectation, at least, was that they would marry. However, after he returned to Nuremberg, his family no doubt wanted nothing to do with an ex-nun and Katharina did not hear from him again.

On 12 October 1524, Luther wrote to Baumgärtner advising, "If you want your Katie von Bora, you had best act quickly, before she is given to someone else who wants her. She has not yet conquered her love for you. I would gladly see you married to each other"[40]. Baumgärtner did not respond, and the rumor traveled to Wittenberg that he had already married another.

As it happened, the other suitor Luther had in mind for Katharina was a certain Doctor Kasper Glatz, the pastor in Orlamünde. Katharina, while perhaps not holding out for love, could not bear the possibility of a future as the wife of Pastor Glatz, who was elderly, and considered to be a miser. As pressure mounted, in September of 1524 Katharina approached Nicholas von Amsdorf for his intervention with Luther. In one of the more charming exchanges in Reformation history, Katharina told Amsdorf that she would prefer (if it could be and it were God's will) to marry Doctor Martin, or to the kindly man with whom she was speaking, Amsdorf himself. Amsdorf, (who never married), agreed with Katharina that she was not suited for Glatz, and after his conversation with Katharina he remonstrated Luther, saying, "What the devil are you intending to do, persuading the good Katy and forcing her?" Luther abandoned his efforts[41].

By mid-May and early June of 1525, following the developments discussed earlier, it was known around Wittenberg that Luther would take Katharina to wife, and the response from his friends was negative, expressing their wish that he marry someone, or anyone, else. In order to forestall the criticism and rumors that had accompanied Melanchthon's and Agricola's earlier engagements[42], and as Frederick had died and the world was ending, Luther then acted quickly. Martin and Katharina were married on 13 June 1525.

Although weddings at the time might take place in or in front of a church, or in the *Hochzeitshaus* (wedding house), with many people participating, it was not the case for Martin and Katharina. They were engaged and married on the same evening in the Augustinian monastery that was Luther's residence, also known as the Black Cloister, in Wittenberg.

The ceremony was attended by a small circle of close and presumably sympathetic friends, including Johannes Bugenhagen who performed the ceremony, Justus Jonas, Mr. and Mrs. Cranach, with whom Katharina was living up until the wedding, and John Apel, a Wittenburg jurist who had

---

38　Ibid., let. 584, p. 181.
39　Ibid., let. 586, p. 183.
40　Ibid., let. 642, p. 258.
41　As reported in Brecht, p. 196. Brecht takes the account from (Kroker 1917).
42　(Brecht 1990, p. 198).

freed and married a nun, a deed for which he had served time in prison. For the small meal with the witnesses that evening the city council of Wittenberg provided seven tankards of Franconian wine, and "thus the city was aware of the event"[43].

Presumably, the order for marriage used was the one Bugenhagen had prepared for Wittenberg in 1524, one that spoke not only of the religious institution of marriage but about the cross that was imposed on the estate of marriage, "a thought with which Luther was very familiar"[44]. Then, according to the local custom, the couple was led into the bedroom where they either "lay down on the marriage bed in front of witnesses", per Brecht, or quite literally consummated the marriage, per Oberman[45]. Whatever happened, we know the witness was Justus Jonas.

The very next day, June 14, Justus Jonas wrote to Spalatin with the news.

> Grace and peace from God. This letter will come to you, my dear Spalatin, as the bearer of great news. Our Luther has married Catharine von Bora. I was present and was a witness of the marriage yesterday. Seeing that sight I had to give way to my feelings and could not refrain from tears. Now that it has happened and is the will of God, I wish this good and true man and beloved father in the Lord much happiness. God is wonderful in His works and ways. . . . There were present Lucas the painter and his wife, Dr. Apel and Bugenhagen. Philip was not there[46].

Speculation varies as to whether the tears Jonas mentioned in his letter to Spalatin were tears brought about by the uncomfortable nature of this particular duty, apprehension about the counter-Reformation backlash, or joy on account of his good friend Luther's happiness.

Two weeks later, on June 27th, Luther held the public procession to the church for a blessing and the *Wirtschaft*, or the marriage feast. Luther's parents and many guests were present. Luther used the intervening two weeks to announce his marriage and to invite friends to the celebration, and also to ask that provisions be donated for the feast[47]. The City of Wittenberg furnished venison and wine for the wedding supper.[48]

After breakfast at the Black Cloister the wedding party processed through Wittenberg to the ringing of church bells to the parish church, had the ceremony at the threshold, proceeded back to the cloister for a dinner, then dancing at the town hall, then another feast at the town hall, and all home by 11:00 PM per the city watch.[49]

Luther's letters of invitation to attend the wedding celebration express surprise that God had so suddenly removed him from bachelorhood and attached him to a woman. And yet, he did not seem unhappy. To friends in Mansfeld he wrote, "So now, according to the wish of my dear father, I have married. I did it quickly lest those praters should stop it" and he included a warm invitation to come to his celebration and housewarming[50]. To Wenzel Link Luther wrote, "Despite the fact that I was otherwise minded, the Lord has suddenly and unexpectedly contracted a marriage for me with

---

43   Ibid., pp. 198–99.
44   Ibid., p. 198.
45   (Oberman 1989, p. 282).
46   (Luther 1918, let. 689, p. 322). Editor Preserved Smith notes, "The first part of this letter is found in a different form", as registered by Spalatin in his *Annals*, ed. Mencken, ii, 645: "Our Luther has married Catharine von Bora. Yesterday I was present and saw the bride lying in the marriage chamber." (Luther 1918, footnote 3, p. 322).
47   Presumably, Leonhard Koppe furnished a barrel of his best beer for the celebration because when Luther wrote on June 21 to invite him, Luther made the request: "Worthy prior and father, God has suddenly and unexpectedly caught me in the bond of holy matrimony. I intend to celebrate the day with a wedding breakfast on Tuesday. That my parents and all good friends may be merry, my Lord Catharine and I kindly beg you to send us, at my cost and as quickly as possible, a barrel of the best Torgau beer. . . . If it is not good, to punish you I shall make you drink it all. I also beg you and your wife not to stay away, but happily to appear". Ibid, let. 694, p. 328.
48   (Bainton 1980, p. 226).
49   Ibid. It is unfortunate that Bainton does not provide references for the description of the events on the day of the public celebration of Luther and von Bora's wedding. He offers more detail than I have found elsewhere.
50   (Luther 1918, let. 690, p. 323).

Catharine von Bora". Luther tells Link that he need not worry about the expense of a gift—just to come to the wedding—and asked for his prayers, as he had to bear many "slanders and reproaches because of this deed that God has done"[51].

In his letter of invitation to Amsdorf, Luther offers a more intimate summary of his thoughts and motivations, writing, "I hope to live a short time yet, to gratify my father, who asked me to marry and leave him descendants; moreover, I would confirm what I have taught by my example, for many are yet afraid even in the present great light of the Gospel. God has willed and caused my act. For I neither love my wife nor burn for her, but esteem her"[52]. Although we can imagine how Luther's thoughts were occupied with the pleasant business of moving his wife into their new home and wedding celebration planning, at the end of the letter Luther revealed that world events were not far from his heart. He concluded, "The poor peasants are perishing by the thousands everywhere. . . . Farewell, and pray for me".

Luther's letter to Spalatin announcing his marriage was happy exultation, if not because of love, then because, by his marriage, he had made the angels laugh and the devils weep:

> Grace and Peace. Dear Spalatin, I have stopped the mouths of my calumniators with Catharine von Bora. If we have a banquet to celebrate the wedding we wish you not only to be present but to help us in case we need game. . . . I have made myself so cheap and despised by this marriage that I expect the angels laugh and the devils weep thereat. The world and its wise men have not yet seen how pious and sacred is marriage, but they consider it impious and devilish in me. It pleases me, however, to have my marriage condemned by those who are ignorant of God. Farewell and pray for me. . . . MARTIN LUTHER.[53]

## 6. Conclusions

Luther's marriage was certainly a confessional act. He had published and preached that marriage was a good and godly thing, needful for all men and women save the very smallest number who had the gift of celibacy; that clergy should marry, that those in holy orders should be released from their vows of celibacy, and he encouraged other prominent reformers and church leaders to marry. It only made sense that the leader of the Reformation should follow his own advice.

The Luther-von Bora marriage exemplified the goals of the Reformation for the reform of marriage and family. The former monk and former nun enjoyed a faithful, productive, marriage in which they appreciated each another physically as well as intellectually, supported one another in good times and bad, loved one another deeply, and raised Christian children. However, in moving within a matter of months from one with no intention of marrying to one who became engaged and married on the same day, other factors came into play.

The fear of impending death as a heretic initially stayed Luther from marrying. However, under some pressure to marry and set an example, and perhaps at some level drawn to the comforts of marriage that he himself so enthusiastically extolled, Luther began to at least consider the idea by 1524. Luther's ongoing discussion with his good friend Spalatin, in which each good-naturedly encouraged the other to marry, shows this. It seems to me that their mutual encouragement must have also moved Luther along.

The death of Frederick the Wise in May of 1525, who did not approve of clerical marriage, removed an impediment to Luther's decision to marry.

Immediately following his marriage, Luther wrote to his family and friends, inviting them to attend the wedding celebration in two weeks' time. In these letters, the explanations Luther offered

---

51    Ibid., let. 694, p. 328.
52    Ibid., let. 695, p. 329.
53    (Luther 1918, let. 691, pp. 323–24).

most frequently for his wedding, were that God had caused it to happen, and that he had married in order to obey and please his father and to stop the mouths of his detractors.

Did he marry for love, or with affection, or anything like it? He wrote to Amsdorf on June 21, 1525, to tell him of his marriage and invite him to the banquet, and added, "For I neither love my wife nor burn for her, but esteem her"[54]. Another interpretation is "cherish". However, it seems likely that there was at least some happy affinity between the former Cistercian nun and the reformer of Wittenberg before they wed. They had known each other since 1523, and the fact that during this time Luther sought suitable situations for all the former nuns in his care, meant that Katharina's wellbeing had been on his mind for two years.

As for Katharina, surely there were other single men in Wittenberg—reformers, professors, pastors—whom Katharina might have married. Katharina was attractive enough in body and spirit to have captured the admiration of the Danish king Christian II, who made her the present of a gold ring, and the heart of the patrician Jerome Baumgärtner. And yet, Katharina named Martin Luther as her only choice for husband, or if failing that, Nicholas von Amsdorf. As events moved Luther quickly towards marriage in the spring of 1525, he may have discovered that Katharina embodied greater virtue and attractiveness than simply being the last remaining nun in his care. Days after the wedding Luther called her "Lord Katie", and within months, writing to Spalatin about his "best little creation of God", surely Luther was also thinking of Katie, "his rib", whom he loved.

Whereas previously, the fear of death as a heretic had held Luther back, the fear of death at the hands of marauding peasants, or as the world itself came to an end, encouraged him to act quickly. While traveling in April of 1525 Luther came to think that the peasants might reach Wittenberg, and just as much, he believed that the violence of the war signaled the end of the world. He believed that the devil, who opposed the spread of the gospel as well as stable, Christian marriages, wished to see him dead by one means or another, and so he married his Katie to spite the devil.

**Funding:** This research received no external funding.

**Conflicts of Interest:** The author declares no conflict of interest.

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
