# Peer review of "To Spite the Devil: Martin Luther and Katharina von Bora’s Wedding as Reform and Resistance"

_religions, doi:10.3390/rel11030116_

Round 1

Reviewer 1 Report

This is a thoughtful and judicious article. The restraint you show in making too many sweeping judgments lends credibility to the conclusion that while he had many personal thoughts about marriage, it was a theological/confessional statement to do so that best explains the reasoning and experience of the event. 

One text that I would think almost necessary to consider is the recent monograph from Natalie Krentz on Ritualwandel und Deutungshoheit. This examination of the first few years of the Reformation in Wittenberg puts Frederick the Wise more at the center than earlier treatments do, but her main focus is on FW's funeral in mid=1525, and how the occasion was a chance to kind of "solidify" in a very public way (e.g., with communion in both kinds offered to all) the basic premises of the Reformation. Your contention that ML's wedding was understood by him to be a confessional act fits with the narrative Krentz construes, but I think it actually would strengthen the flipside of your contention, namely that ML's wedding was understood by others to be a confessional act. 

A couple little things: Heiko, not Haiko in fn. 12, journal title is underlined in fn 2 but not in fn 4 (check that throughout). Ozment's work on family life in EME has been strongly criticized by feminist scholars, notably Merry Wiesner-Hanks (see her article on marriage in the recent Oxford Encyclopedia of Martin Luther), but if you're using him for factual/data citations that's probably fine. 

In re: the Peasant's War, ML's take on how that event unfolded is entirely wrapped up in the publication history. His many writings did not come out in the order in which he wrote them, and so he always felt misrepresented. 

Yes, it was an apocalyptic event for him, but i think you (following HO) perhaps make too much of that. Much of his talk about the devil coming back to try to reign are just figures of speech that were commonly used then. 

Lastly, one suggestion would be to look at some other texts of ML's than the usual suspects. He preached several sermons at weddings around this time, and he often indicates subtle features of his theology of marriage (as well as some occasionally 'personal' ones). E.g. his sermon at Lazarus Spengler's wedding. Same at Link's wedding. 

Author Response

Thank you for your kind comments and thoughtful review. I’m appreciative.

Regarding Natalie Krentz’s work, Ritualwandel und Deutungshoheit, thank you for pointing me to this work. It’s fascinating and important, and I’ll cite it as you suggest. Regarding the use of Ozment on family life, it is not my intention in this article to assess from a historian or theologian’s perspective the effect of the Reformation on women’s lives, positive, negative, or a mixed bag. Rather, I present Luther’s perspective on the state of marriage at the time to illustrate why he and the other reformers believed marriage needed to be reformed, which speaks to the question of why he married when he did (and why hadn’t he married before then as he had encouraged others to do?). I use Ozment to create a larger social picture. I’ll include a brief discussion or long footnote, though, on the question of how the Reformation brought about changes, gains, and losses, for women, knowing of course, that it all depends… I’ll make the spelling and other corrections. Regarding Luther and the devil. I have been pondering this, because yes, I agree, many of Luther’s mentions of the devil were figures of speech or common parlance, but he did also live in a time in which the devil’s power and influence was considered real and influential in the world. When he says, for example, that he married to spite the devil, this is both figurative and literal. But perhaps I am overly influenced by Oberman… Luther’s apocalyptic fears are entangled with his general fear of and loathing for violence and social unrest. A kind of “the world is ending” sensibility is present, whether because the Lord would return or because the marauding peasants would injure him or destroy his town. I appreciate the suggestions to read more marriage sermons. This may be part of future research. My research style preference is to do close readings of historical texts, and so this would work very well.

Reviewer 2 Report

The paper has an engaging and well-written narrative. The paper was enjoyable to read. My greatest reservation, however, lies in the paper's covering of well-trod ground drawn from thoroughly picked-over materials. 

The paper is a classic intellectual history utilizing translated primary source materials from Martin Luther, with contemporary supporting materials from Philipp Melanchthon and Spalatin. Most of the findings of the paper are drawn from classics in Lutheran historical studies: Oberman, Lull, Brecht, and Bainton -- much of these nearly 30 years old. The paper offers interesting new opportunities for insights at the intersections of class and theology in the marriage narrative and places for new interpretations regarding the Peasant War and Luther's own participation in the conflict. Luther a successful Burgher and von Bora, a member of the nobility's concerns about the real contemporaneous sufferings of the Peasants is not fully explored, however, as the paper maintains close connection to the sources from Luther rather than interacting with some of the social historical interpretations and sources of the sixteenth century. I would recommend integrating these aspects into the work.

Author Response

Thank you for your comments on my paper. I appreciate your kind comments about its readability, as well as some of the reservations you express. One reason for using, as you say, well-worn resources is that these are the wells scholars return to in any discussion of the Luther/von Bora marriage, and I was interested in not simply scooping out a familiar phrase here and there but returning to the work and laying out its perspective and details more thoroughly, especially for an audience that is perhaps unlikely to do it themselves. I appreciate your pointing out that there are opportunities for insights into the intersection of class and theology as related to marriage and Luther’s marriage, but I think this would be a different paper.

Reviewer 3 Report

This paper is very well written and well documented. Above all, it is interesting—indeed fascinating—to read. The authors take us through the sequence of events that led Luther to marry Katharina von Bora. They detail the role that Luther's theological and apocalyptic beliefs, as well as chance historical events (such as the death of Elector Frederic the Wise), played in leading him to a decision that immediately stressed his relationship with his supporters. The portraits of both Luther and Katharina are well developed. While I am not a Luther scholar, I am familiar enough with Luther's theology and writings on marriage to say that the total account offered here is sound. The summary finding, reflected in the title, that Luther acted to "spite the devil" is original and compelling.

Two very minor suggestions for improvement:

On p. 4, lines 162-3, the authors introduce a letter dated April 16, 1525. But it is not clear until after the letter is quoted just whose letter it is, Luther's or Spalatin's. I would change "In a letter dated April 16, 1525,..." to "In a letter written by Luther dated April 16, 1525,..." 

On p. 6, lines 252-3, the statement "the devil hated the joys of marriage" could use a supporting citation. 

Author Response

Thank you for your kind comments – they were received with pleasure. I will make the corrections you suggest.

Round 2

Reviewer 2 Report

The paper still does not integrate or utilize new interpretations or understandings of the relationship between Luther and von Bora. I do not believe that this paper offers the readers of Religions a new understanding of gender roles or the ideology of Luther. As mentioned in the original critiques, I do believe that there are interesting intersections with gender, theology, and class. However, they are not fleshed out or explored here.

Author Response

I am appreciative of the reader's time and suggestions. I agree that there are many interesting intersections that could be explored, and are not explored, in this paper. However, the question being pursued is, "why did Luther choose to marry when he did?" and whereas there is new social data about marriage and families and gender relations during the time of the Reformation, in fact there is very little information available that describes precisely, specifically, Luther's thoughts and motivations about his marriage before he married (his letters are filled with references afterward). As I'm sure the reader knows, all but a handful of von Bora's letters are lost, and none that were written to ML. Similarly, there is very little to be found that describes the wedding and the feast that followed two weeks later. For this, most scholars return to the same sources, and in this paper I have done that as well. However, in answering the question posed I have argued for a number of influences -- theological, personal/familial, political, and confessional -- and the confluence of motivations which I identify and for which I argue is original and is a contribution to readers of Religions